# MOSLIM: ALIGN WITH DIVERSE PREFERENCES IN PROMPTS THROUGH REWARD CLASSIFICATION

## ABSTRACT

The multi-objective alignment of Large Language Models (LLMs) is essential for ensuring foundational models conform to diverse human preferences. Current research in this field typically involves either multiple policies or multiple reward models customized for various preferences, or the need to train a preference-specific supervised fine-tuning (SFT) model. In this work, we introduce a novel multi-objective alignment method, MOSLIM, which utilizes a single reward model and policy model to address diverse objectives. MOSLIM provides a flexible way to control these objectives through prompting and does not require preference training during SFT phase, allowing thousands of off-the-shelf models to be directly utilized within this training framework. MOSLIM leverages a multi-head reward model that classifies question-answer pairs instead of scoring them and then optimize policy model with a scalar reward derived from a mapping function that converts classification results from reward model into reward scores. We demonstrate the efficacy of our proposed method across several multi-objective benchmarks and conduct ablation studies on various reward model sizes and policy optimization methods. The MOSLIM method outperforms current multi-objective approaches in most results while requiring significantly fewer GPU computing resources compared with existing policy optimization methods.

## 1 INTRODUCTION

While large language models (LLMs) have been widely adopted across various domains, generating text that aligns with human preferences has become a prominent area of research. Stiennon et al. (2020) introduced the concept of learning from human feedback to better align model behavior with human preferences, specifically aiming to produce summaries that are more preferred by human annotators. Ouyang et al. (2022) proposed the Reinforcement Learning from Human Feedback (RLHF) approach within InstructGPT framework, employing a combination of supervised fine-tuning and reinforcement learning to align models with human-defined preferences, such as instruction-following and safety. Later that year, Bai et al. (2022) also proposed an alignment method based on human feedback, focusing on enhancing helpfulness, harmlessness, and honesty.

However, it has become evident that models aligned with mixed general preferences often struggle to address diverse needs of different application scenarios (Zhang 2023; Lee et al. 2024; Kirk et al. 2024). The preferences required for varying tasks and users can differ significantly. For example, in certain cases, it may be necessary to prioritize helpfulness over honesty, thereby accepting a degree of hallucination. Consequently, multi-objective preference alignment has emerged as a significant area of research. Researchers have begun exploring methods to apply different combinations of preferences to generate content that better suits specific contextual requirements (Ramé et al. 2023; Zhou et al. 2023; Jang et al. 2023; Yang et al. 2024; Guo et al. 2024b; Wang et al. 2024a; Li et al. 2024d). Methods such as MORLHF, Rewarded Soups(Ramé et al. 2023), and MODPO(Zhou et al. 2023) have been proposed to address this challenge by training models to align with distinct preference combinations. However, techniques like MORLHF and MODPO can only align with a specific combination of preferences once training is completed, whereas reward soups allows for more flexibility by combining preferences at the inference stage but at the cost of increased computational and training overheads due to the need to train multiple policies.

To address these limitations, researchers have explored methods for dynamically adjusting content generation preferences through the use of prompts. Yang et al. (2024) proposed the RiC method, which leverages supervised fine-tuning (SFT) to enable models to respond according to different tags embedded in the prompts. Guo et al. (2024b) introduced the CDPO approach, asserting that most real-world applications require balancing only a limited set of preferences simultaneously. These methods rely primarily on SFT phase, with some incorporating additional RLHF training. However, under the current LLM training paradigm, the SFT stage is mainly responsible for enhancing core capabilities such as mathematical reasoning, coding, and specialized domain knowledge, while the RLHF stage is used to integrate human-aligned responseset al (2024); Lu et al. (2024). As a result, directly incorporating complex preference combinations and intensities during SFT stage is impractical for real application scenarios. For instance, defining what constitutes a *helpful*, *harmless*, or *honest* response in a mathematical context is inherently challenging.

In this paper, we propose the **MOSLIM** method, which enables content generation aligned with varying combinations of preferences using prompt-based control while significantly improving training efficiency. MOSLIM eliminates the need to incorporate preference-specific generation capabilities during SFT phase, achieving dynamic preference alignment solely through the RLHF stage. This allows off-the-shelf models to be directly integrated with MOSLIM without requiring substantial modifications or retraining. MOSLIM employs a multi-head classification reward model and a policy gradient approach based on a reward mapping function to align with any combination of preference objectives and intensities using a single reward model and policy model during training. During inference, a prompt-driven mechanism enables flexible adaptation to varying preference intensity combinations.

We validate our method through experiments using three distinct preference evaluation benchmarks, demonstrating its effectiveness in achieving controllable preference alignment. Furthermore, we assess performance across different intensity levels, highlighting significant behavioral variations as the intensity of preferences changes. To the best of our knowledge, our approach is the first to achieve dynamic preference alignment using a single reward model and policy while with the ability of leveraging off-the-shelf models, representing a novel contribution to the field of multi-objective preference alignment.

## 2 METHODOLOGY

### 2.1 PRELIMINARIES

The multi-objective alignment begins by extending RLHF (Reinforcement Learning from Human Feedback) framework to meet users' varying combinations of preferences, commonly referred to as MORLHF. Rather than focusing on a single objective or reward model, this approach trains multiple reward models, each corresponding to a different preference. During policy optimization phase, the rewards from these models are combined through a weighted summation, as shown in Eq 1:

$$R = w_1 r_1 + w_2 r_2 + \cdots + w_n r_n \tag{1}$$

Here, $r_i$ represents the reward for the $i$-th preference, while $w_i$ denotes the weight assigned to that preference for a specific task or user. This weighted sum helps guide the model toward achieving a balanced alignment of user preferences.

However, MORLHF requires retraining for each combination of preferences. To address this, Rewarded Soups (Ramé et al. 2023) proposes a novel method for parameter combination. It involves training separate reward models and policy models for each individual preference. During inference, policy parameters are combined based on the specific preference requirements of the scenario, producing tailored results. The formulation is shown in Eq 2:

$$\theta_\lambda = \sum_{i=1}^{N} \lambda_i \cdot \theta_i \tag{2}$$

where $\theta_i$ represents the policy parameters trained exclusively for the $i$-th preference and $\lambda_i$ is the weight for $i$-th preference.

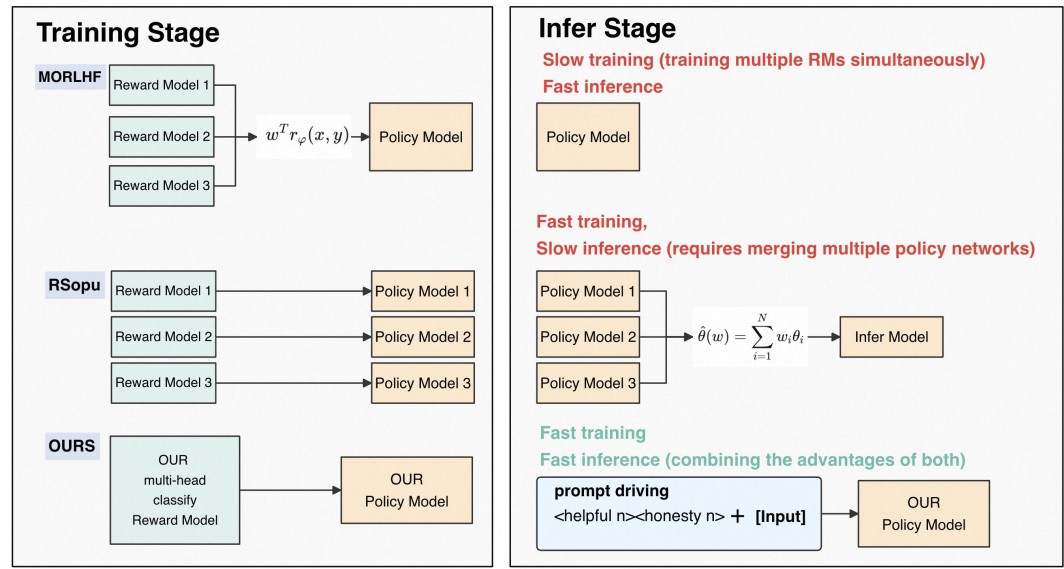

Figure 1: An overview of MOSLIM(OURS), MORLHF, Rewarded Soups(RSopu) during training and infer stages.

Both MORLHF and Rewarded Soups focus on multi-objective modeling based solely on preference dimensions. However, recent works (Yang et al. 2024; Guo et al. 2024b) has extended these approaches by modeling the intensity of preferences within the same dimension and optimizing for preference-specific content at varying levels of intensity. The loss function for Conditional Direct Preference Optimization (CDPO, Guo et al. (2024b)) is shown in Eq 3:

$$\mathcal{L}_{\text{CDPO}} = -\mathbb{E}_{(x,c,y_w,y_l)\sim\mathcal{D}} \left[ \log \sigma \left( \hat{R}_\theta(c, x, y_w) - \hat{R}_\theta(c, x, y_l) \right) \right], \tag{3}$$

where $\hat{R}_\theta(c, x, y_w) = \beta \log \frac{\pi_\theta(y_w|c,x)}{\pi_{\text{ref}}(y_w|c,x)}$ and $\hat{R}_\theta(c, x, y_l) = \beta \log \frac{\pi_\theta(y_l|c,x)}{\pi_{\text{ref}}(y_l|c,x)}$ represent the implicit rewards in the DPO (Rafailov et al. 2023) algorithm. Here, $c$ refers to the control tag in prompt, which includes both preference and intensity, such as `<helpfulness 5>`.

Our method, MOSLIM, combines the training methodologies of multi-objective policy optimization with multi-dimensional intensity control techniques. MOSLIM optimizes across various dimensions of user preferences and their corresponding intensities. Notably, we achieve alignment across all dimensions and intensities using a single reward model and policy model as showed in Figure 1. We will elaborate on our methodology in the subsequent sections.

## 2.2 REWARD MODELING

In the context of Large Language Models (LLMs), a reward model is utilized to assess quality of the outputs generated by these models. The original formulation of reward model modifies final output layer of LLMs to transform token probability distribution into a single score (Stiennon et al. 2020), as illustrated in Eq 4.

$$r = f(x, y) \tag{4}$$

In this equation, $x$ represents input prompt, while $y$ denotes the corresponding answer.

As research on preference alignment evolved, this training paradigm for reward models is found to obscure significant preference information by adhering solely to a majority voting principle, thereby overlooking the nuances between majority and minority groups (Jang et al. 2023). Li et al. (2024a) introduced a distribution-based reward model that outputs a distribution of preferences, as described in Eq 5.

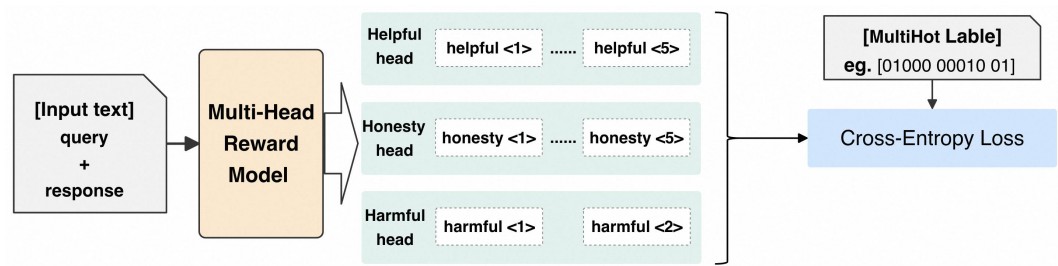

Figure 2: Reward Model Architecture of MOSLIM

$$P^G(x,y) = \left[ l_j^G(x,y) \right]_{j=1}^d = \left[ \frac{\sum_{u_i \in G} l_1^{u_i}(x,y)}{|G|}, \ldots, \frac{\sum_{u_i \in G} l_d^{u_i}(x,y)}{|G|} \right] \tag{5}$$

In this formula, $G$ denotes the total number of annotators, and $l_d^{u_i}(x,y)$ represents the preference of annotator $u_i$ for the $d^{th}$ preference dimension.

Inspired by distribution modeling, we propose a multi-headed reward model that utilizes a single reward model to capture multiple distinct preference objectives. We modify the output layer of reward model from a single score to multiple preference heads. We categorize a question-answer (Q,A) sequence into preferences such as helpfulness, harmlessness, or honesty. Within each head (e.g., helpful), we classify the intensity of the preference. We then optimize our reward model by calculating the accuracy of intensity classification for each head. The overall architecture is showed in 2. We calculate classification accuracy for each head individually, as expressed in Eq 6, where $S_i$ refers to the softmax output of the $i^{th}$ preference head, and $L_i$ denotes the label for $head_i$.

$$L_{R_{head_i}} = CrossEntropy(S_i, L_i) \tag{6}$$

The softmax output $S_i$ is defined as follows:

$$S_i = \text{Softmax}(z_i) = \left[ \frac{e^{z_{i,j}}}{\sum_{k=1}^K e^{z_{i,k}}} \right]_{j=1}^K \tag{7}$$

where $z_i$ is the vector of logits corresponding to the $i^{th}$ preference head, $K$ represents the total number of classes, and $j$ indexes each class.

The label $L_i$ for the $i^{th}$ preference head can be represented as a one-hot encoded vector:

$$L_i = \begin{cases} 1 & \text{if class} = y \\ 0 & \text{otherwise} \end{cases} \tag{8}$$

where $y$ is the true class.

The Cross-Entropy loss $CrossEntropy$ measures the difference between the true labels and the predicted probabilities. For the $i^{th}$ preference head, it is defined as:

$$L_{R_{head_i}} = CrossEntropy(S_i, L_i) = -\sum_{j=1}^K L_{i,j} \cdot \log(S_{i,j}) \tag{9}$$

where $L_{i,j}$ is the $j^{th}$ element of the one-hot encoded label $L_i$, and $S_{i,j}$ is the predicted probability for class $j$.

For computational convenience, we aggregate the losses from all heads using the following equation:

$$L_R = CrossEntropy(S_1||S_2||\dots||S_n, L_1||L_2||\dots||L_n)$$

$$= -\sum_{j=1}^{K} L_j \cdot \log\left(\frac{e^{z_{1,j}} + e^{z_{2,j}} + \dots + e^{z_{n,j}}}{\sum_{k=1}^{K}(e^{z_{1,k}} + e^{z_{2,k}} + \dots + e^{z_{n,k}})}\right) \tag{10}$$

where $S_1, S_2, \dots, S_n$ are the softmax outputs from different heads, and $L_1, L_2, \dots, L_n$ are their corresponding labels.

## 2.3 POLICY OPTIMIZATION

In this section, we are aiming to train policy models using the reward model constructed in 2.2. The required data format in this stage consists of prompt data with specific preference dimensions and intensities, which should be consistent with the objectives defined in reward model training. An example of training samples used in this phase is shown in Appendix A. The entire policy optimization method involves two key components: **prompt alignment** and **reward mapping**, with a complete policy optimization process depicted in Appendix A.

**Prompt Alignment**: Since prompts did not include preference labels during reward model training, we need to remove these labels when obtaining reward scores in the policy optimization stage. This adjustment ensures that inference data received by reward model remains consistent with the data format used during its training. The data flow for both the reward model and policy model inputs is illustrated in Appendix A.

**Reward Mapping**: Most current policy optimization methods receive a scalar reward signal. Therefore, we need to apply a reward mapping strategy for multi-head reward model to aggregate preference classification results into a unified scalar. The reward mapping process addresses the following two main challenges:

1. **Dimensional Variability**: The number of preference dimensions may vary depending on the specific business scenario, requiring compatibility with any number of preference dimensions.

2. **Intensity Scaling**: Each preference dimension may have different levels of controllability. For example, the *helpfulness* dimension might have 5 levels, while the *harmless* dimension might only have 2. Consequently, scores across different dimensions need to be scaled to a consistent metric for comparability.

To address these challenges, we propose a reward mapping function, which can scale both dimensions and intensities of preferences. During the training phase, we record the moving average and standard deviation of each preference intensity within every preference head. During inference phase, each preference dimension value is transformed into a sample value from a Gaussian distribution with zero mean and unit variance, making the intensity scores from different preference dimensions additive. The specific reward mapping formula is defined as follows:

$$r_{\text{score}} = \frac{1}{k} \sum_{i=0}^{k} \frac{(p_i^{\text{target}} - p_i^{\text{avg}})}{p_i^{\text{std}}} mask_i \tag{11}$$

where $i$ denotes the preference dimension, $k$ represents the total number of preference dimensions, and $target$ represents preference intensity for $i$-th dimension in the prompt. Notably, if a preference dimension is not specified in the prompt, $mask_i$ will be set to 0, other wise, $mask_i$ equals to 1.

With the resulting reward mapping, we can perform policy optimization. Our framework supports various policy optimization algorithms, like Proximal Policy Optimization (PPO) [Schulman et al. 2017], Reinforcement Learning from Optimal Outcomes (RLOO) [Ahmadian et al. 2024] , and Online Direct Preference Optimization (Online DPO) [Guo et al. 2024a]. The following equation illustrates a Policy Optimization formula used for PPO:

$$\text{objective}(\phi) = \mathbb{E}_{(x,y)\sim\mathcal{D}_{\pi_\phi^{\text{RL}}}}\left[r_\theta(x,y) - \beta \log\left(\frac{\pi_\phi^{\text{RL}}(y\mid x)}{\pi^{\text{SFT}}(y\mid x)}\right)\right] \tag{12}$$

where $\pi_\phi^{\text{RL}}$ denotes the learned RL policy, $\pi^{\text{SFT}}$ refers to the supervised fine-tuned model. The KL divergence coefficient, $\beta$ controls the strength of the KL penalty.

Through the two stages of reward modeling and policy optimization, we obtain a complete training framework, which we refer to as **MOSLIM**. This scheme is capable of training any combination of preference dimensions and intensities using a single reward model and a single policy model. Moreover, it enables flexible control during inference.

## 3 EXPERIMENT

In this section, we are aiming to answer the following questions:

1. **Effectiveness of the Reward Model**: Does the proposed reward model paradigm effectively classify tasks and conform to the scaling law?

2. **Superiority of MOSLIM-trained Policies**: Do policies trained with MOSLIM demonstrate clear advantages in specific preference dimensions and intensities?

3. **Impact of Preferences**: How do the number of combined preference dimensions and varying intensities affect the model's overall performance?

### 3.1 REWARD MODEL

We conduct a series of experiments to evaluate the effectiveness of the proposed reward model. First, we train the reward models with varying parameter sizes and assess their classification accuracy across three different scales: 7B, 57B, and 72B, corresponding to there SFT models trained with hh-full-rlhf (Guo et al. 2024c) dateset based on Qwen2-7B-base, Qwen2-57B-base, and Qwen2-72B-base models, respectively. This experiment aims to varify weather multi-head classification pattern reward model works and is scaling law still consist when turning into a classification pattern. Detailed hyperparameters used for training the reward models are provided in Appendix D,Table 7. Next, we conduct ablation studies on datasets with varying numbers of classification categories. All three reward models are tested on four datasets with different category distributions. Finally, we compare accuracy between our trained reward model with GPT-4, which acting as a llm annotator, demonstrating that our reward model achieves significant performance gains over GPT-4 annotators. The detailed results compared with GPT-4 is showed in Appendix E,Figure 9.

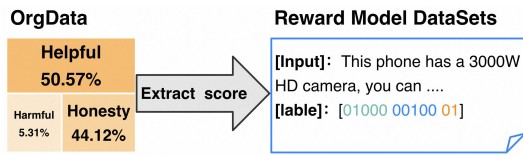

Figure 3: Construction process of reward model training datasets.

**Data Construction:** We utilize two open-sourced datasets to train our reward model:UltraFeedback (Cui et al. 2024) and UltraSafety (Guo et al. 2024c). UltraFeedback dataset contains 64,967 high-quality preference data points along with corresponding preference score labels. We extract the *helpfulness* and *honesty* preference data from this dataset. UltraSafety dataset includes 3,000 carefully curated harmful instructions and annotations, providing training data for the *harmless* preference dimension. We construct training data for our reward model by mapping dimention scores to preference intensities. Each sample in our reward datasets primarily consists with *input* and *label*. *Input* includes a question and its corresponding answer, while *label* is a multi-hot classification vector, representing the specific preference dimensions and intensities, as illustrated in Figure 3. Totally four types of training data (*DataType* 1-4) are generated based on different score mapping schemes. The larger value of $n$ (the number of data types) indicates more numbers of classification categories. Detailed definitions of *DataType* 1-4 are shown in Appendix B.

**Performance Evaluation:**We conducted experiments to evaluate the performance of our multi-head reward model, testing it on three different model sizes as mentioned previously: *RM-7B*, *RM-57B*, and *RM-72B*. The accuracy results are summarized in Table 1. All preference accuracies exceed

Table 1: Model accuracy across different preference categories and intensities for various data types.

| Model | DataType 1 | | DataType 2 | | DataType 3 | | DataType 4 | |
|---|---|---|---|---|---|---|---|---|
| | Preference | Intensity | Preference | Intensity | Preference | Intensity | Preference | Intensity |
| **RM-7B** | 0.9204 | 0.9169 | 0.8898 | 0.6598 | 0.8634 | 0.4734 | 0.8710 | 0.2910 |
| **RM-57B** | 0.9421 | 0.9393 | 0.8919 | 0.7066 | 0.8731 | 0.5153 | 0.8876 | 0.3635 |
| **RM-72B** | **0.9692** | **0.9649** | **0.9398** | **0.7219** | **0.9134** | **0.5491** | **0.8910** | **0.3824** |

87%, and the intensity classification results for both *Datatype* 1 and *Datatype* 2 are above 65%. Moreover, as the model size increases, performance improves significantly. For instance, when comparing *RM-7B* with *RM-72B* on the *DataType* 4 dataset, we observe a notable accuracy improvement of nearly 10%. On the simpler *DataType* 1 dataset, however, the performance gain is relatively modest, around 5%. These results suggest that the advantages of larger reward models become more pronounced as the complexity of the dataset increases, thereby significantly enhancing classification accuracy for more challenging tasks. Figure 4 further confirms this relationship between model performance and dataset complexity: as the granularity of the *DataType* increases, model accuracy declines. For example, with *DataType* 4, even the *RM-72B* achieves an accuracy of only 0.38, which is approximately 58.25% lower than its performance on *DataType* 1. Nonetheless, reward models trained on more complex data, such as *DataType* 4, still demonstrate effectiveness and contribute to performance gains in RL training scenarios.

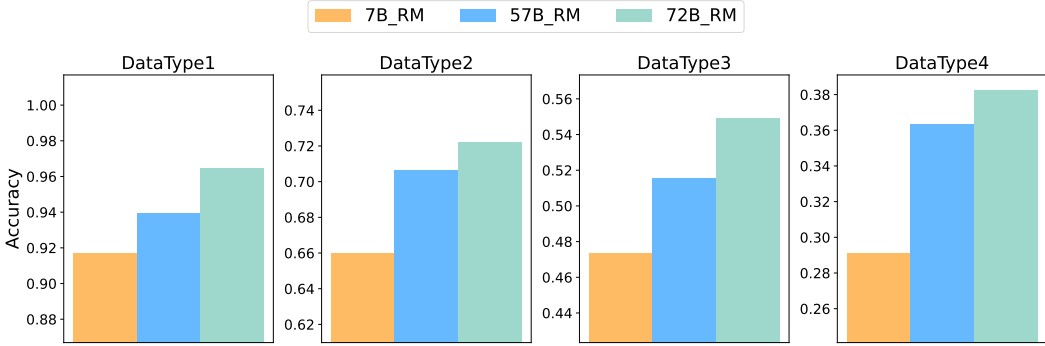

Figure 4: Ablation study on different *DataType* . The figure illustrates the classification performance across four *DataType* .

## 3.2 POLICY OPTIMIZATION

In this section we conduct our policy optimization experiments with the reward models trained in 3.1. All policy models are initialized from the same 7B SFT model that is used to train the reward model. We compare our MOSLIM with baseline methods, MORLHF and Rewarded Soups (RSoup), to demonstrate the effectiveness of our training paradigm. All three policy models are trained using PPO as policy optimization algorithm. The training hyperparameters used for these methods can be found in Appendix D, Table 6. Additionally, we conduct ablation studies on different sizes of reward models: *RM-7B*, *RM-57B*, and *RM-72B*, to investigate the impact of reward model size on training performance. We further validate the generality and effectiveness of our approach using both RLOO and Online-DPO as policy optimization methods.

**Policy Optimization Data Construction:** We first construct a policy optimization training dataset with 34K prompts sampled from the open-source dataset full-hh-rlhf,excluding the data used for SFT training.We extract prompts from the full-hh-rlhf dataset and augment them with preference dimensions, which consist of two levels: preference dimensions and preference intensities. The preference dimensions are assigned randomly from *helpfulness*, *harmlessness*, *honesty*. We randomly select 1-3 preference dimensions. Preference intensities are also randomly assigned. In *helpfulness* and *honesty*, it is ranging from 1-5. For *harmlessness*, it is ranging from 0-1. The whole flow of constructing the dataset is illustrated in Figure 8. Each sample in dataset consists of two parts: a preference prefix

and a prompt. The preference prefix is formatted as `<preference n>`, where `<preference >` denotes the desired preference dimension, and `n` indicates the preference intensity. The prefix explicitly sets the target of the current task, while the prompt contains the actual task input. The model is required to generate responses that align with the preference dimensions and intensities specified by the prefix.

We evaluate our policy models with three benchmarks: MT-Bench (Zheng et al. (2023)), HaluEval 2.0 (Li et al. (2024b)), and Hackaprompt (Schulhoff et al. (2024)). The models are assessed across three preference dimensions: *helpfulness*, *honesty*, and *harmless*.

Table 2: Comparison of (PPO RSoup MORLHF) across different DataTypes.

| Method | DataType 1 | | | DataType 2 | | | DataType 3 | | | DataType 4 | | |
|---|---|---|---|---|---|---|---|---|---|---|---|---|
| | Helpful | Honesty | Harmless | Helpful | Honesty | Harmless | Helpful | Honesty | Harmless | Helpful | Honesty | Harmless |
| **MORLHF** | 2.22 | 2.65 | 0.72 | 2.22 | 2.65 | 0.72 | 2.22 | 2.65 | 0.72 | 2.22 | 2.65 | 0.72 |
| **RSopu** | 2.84 | 3.01 | 0.77 | 2.84 | 3.01 | 0.77 | 2.84 | 3.01 | 0.77 | 2.84 | 3.01 | 0.77 |
| **MOSLIM** | **3.54** | **3.16** | **0.79** | **3.36** | **3.25** | **0.81** | **3.36** | **3.13** | **0.78** | **3.14** | **3.11** | **0.81** |

**MOSLIM vs. Baselines:** We compare the performance of our MOSLIM method with baseline methods MORLHF and RSoup. All policies are train with 7B sized reward model. As shown in Table 2, the evaluation is conducted on four datasets of varying difficulties. The results demonstrate that MOSLIM method achieves superior performance across all preference dimensions Notably, in the most challenging dataset, *DataType* 4, MOSLIM outperforms the RSoup method by 24% in terms of helpfulness scores. These findings confirm the effectiveness of our approach, as even the smallest MOSLIM model surpasses the baseline methods in performance while being more efficient in terms of training time. Among the baselines, MORLHF performs the worst, while Rewarded Soups shows slightly better overall results than MORLHF. We also compare the GPU training time to show our efficacy. MOSLIM requires significantly less GPU time compared to the baselines, as listed in Appendix E, Table 8).

Table 3: Comparison of MOSLIM with three types of RMs across different DataTypes.

| Method | DataType 1 | | | DataType 2 | | | DataType 3 | | | DataType 4 | | |
|---|---|---|---|---|---|---|---|---|---|---|---|---|
| | Helpful | Honesty | Harmless | Helpful | Honesty | Harmless | Helpful | Honesty | Harmless | Helpful | Honesty | Harmless |
| **PPO$_{RM-7B}$** | 3.54 | 3.16 | 0.79 | 3.36 | 3.25 | 0.81 | 3.36 | **3.13** | 0.78 | 3.14 | 3.11 | 0.81 |
| **PPO$_{RM-57B}$** | 3.59 | 3.35 | 0.83 | 3.42 | 3.33 | 0.84 | 3.35 | 3.01 | 0.77 | 3.22 | 3.12 | **0.87** |
| **PPO$_{RM-72B}$** | **3.63** | **3.40** | **0.85** | **3.51** | **3.41** | **0.92** | **3.40** | 3.11 | **0.89** | **3.29** | **3.14** | 0.85 |

**Reward Model Size Scaling in MOSLIM:** We conduct ablation studies on reward model size on MOSLIM. We use reward models of three different sizes we trained in 3.1:7B, 57B, and 72B as the reward and critic models in PPO training. The results, revealed in Table 3, show a positive correlation between reward model size and performance, like PPO$_{RM-72B}$ achieves up to 0.2 points higher helpfulness scores compared to PPO$_{RM-7B}$, indicating the presence of a reward model scaling law in policy optimization phase.

Table 4: Comparison of MOSLIM with different algorithms across different DataTypes.

| Method | DataType 1 | | | DataType 2 | | | DataType 3 | | | DataType 4 | | |
|---|---|---|---|---|---|---|---|---|---|---|---|---|
| | Helpful | Honesty | Harmless | Helpful | Honesty | Harmless | Helpful | Honesty | Harmless | Helpful | Honesty | Harmless |
| **PPO$_{7B-RM}$** | 3.54 | 3.16 | 0.79 | 3.36 | 3.25 | 0.81 | 3.36 | 3.13 | 0.78 | 3.14 | 3.11 | 0.81 |
| **RLOO$_{7B-RM}$** | 3.49 | 3.37 | 0.88 | 3.39 | 3.35 | 0.87 | 3.37 | 3.10 | 0.83 | **3.37** | 3.21 | 0.82 |
| **Online-DPO$_{7B-RM}$** | **3.82** | **3.51** | **0.91** | **3.74** | **3.52** | **0.88** | **3.47** | **3.14** | **0.86** | 3.32 | **3.25** | **0.86** |

**Comparison of Different Policy Optimization Methods:** In this part, We compares MOSLIM performance with three policy optimization methods: PPO, RLOO, and Online-DPO. All methods are trained with 7B SFT model and *RM-7B*. The results, as shown in Table 4, demonstrate significant

performance differences among these methods. RLOO outperforms PPO with up to a 7% improvement in *helpfulness* and also shows better performance in the dimensions of *honesty* and *harmless* . On *DataType* 3 dataset, RLOO achieves a score of 3.37 in *honesty* , compared to PPO's 3.16, representing an improvement of approximately 6.6%. Online-DPO performs best across most preference dimensions and data types. On the *DataType* 3 dataset, Online-DPO achieves 3.82 in *helpfulness* , significantly higher than PPO's 3.54 and RLOO's 3.49. In the *honesty* dimension, Online-DPO scores 3.51, surpassing both PPO's 3.16 and RLOO's 3.37, demonstrating its superior performance.

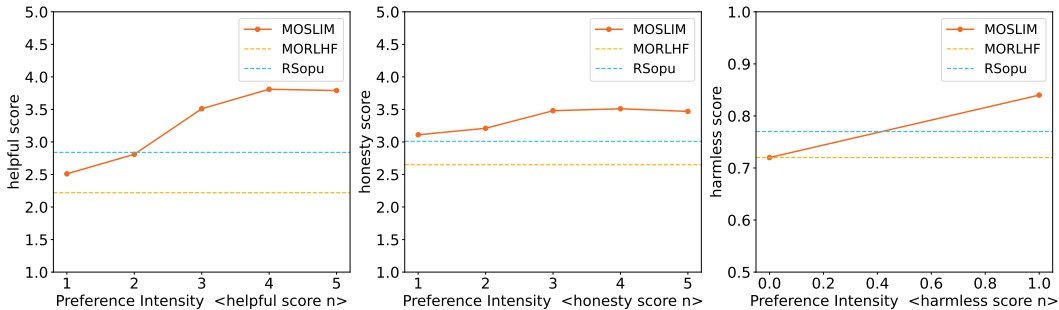

Figure 5: Controllability experiment results of preference intensity. From left to right, the subfigures represent preference goals `<helpfulness n>`, `<honesty n>`, and `<harmless n>`, with the y-axis indicating the preference evaluation scores of the model outputs. As the preference intensity $n$ increases, the scores exhibit a clear upward trend.

**Controllability of Preference Intensity** To validate that model outputs are influenced by preference intensity, we conduct a controllability experiment using Online-DPO model trained on *DataType* 4 dataset. We set a single preference goal, `<preference n>`, and then vary the preference intensity value $n$. The results are shown in Figure 5, demonstrating the controllability of our method's intensity. The x-axis represents the set preference intensity values $n$, and the y-axis shows the preference evaluation scores for the model outputs. As preference intensity increases, MOSLIM model's preference score exhibits a clear upward trend, especially in the *helpfulness* dimension. As the intensity value $n$ increases from 1 to 5, the *helpfulness* score rises from 2.5 to 3.8. Similarly, the *honesty* score increases from 2.51 to 3.79, reflecting a difference of 1.28. In the *harmless* dimension, the score rises from 0.72 to 0.77. The baseline methods MORLHF and RSoup do not have the ability of awaring the preference indensities.

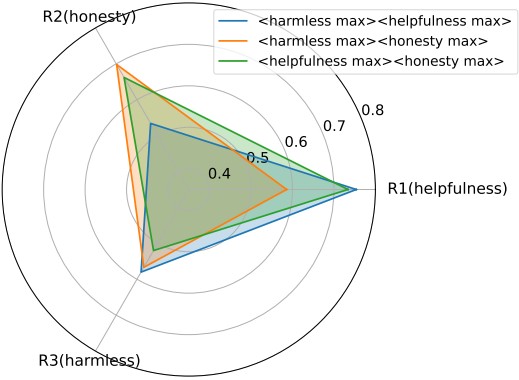

Figure 6: Controllability experiment results across preference dimensions. The scores in each dimension show the model's ability to balance and control performance across different combinations of preference goals.

**Controllability of Preference Dimensions** To demonstrate the controllability of our model across different preference dimensions, we conduct a dual-objective preference experiment. Specifically, we evaluate the model's performance by pairing all possible combinations of preference goals and measuring the preference scores in three dimensions. The results are presented in the radar charts in

Figure 6, using the Online-DPO model trained on the *DataType* 4 dataset for evaluation. When the preference prefix is set to `<harmless max><helpfulness max>`, the model achieves scores of 0.87 in harmlessness and 3.77 in helpfulness, which are significantly higher than those observed for other preference prefixes. Similarly, by adjusting the target preferences, we observe that model consistently performs better on specified preference objectives. This demonstrates the controllability of MOSLIM approach across different preference dimensions.

## 4 RELATED WORKS

**Multi-objective Alignment**: Multi-objective alignment approaches aim to extend RLHF (Ouyang et al., 2022) by accommodating multiple preference objectives simultaneously. One foundational method is MORLHF, which trains separate reward models for each preference dimension. *Rewarded Soups* (Ramé et al., 2023) expand this framework by training multiple policies and merging their parameters, enabling flexible preference combinations at inference time. Similarly, Wang et al. (2024b) leverage parameter merging within a multi-task paradigm to train a conditioned LLM capable of handling diverse preferences. Park et al. (2024) propose a clustered Direct Policy Optimization (DPO, Rafailov et al. 2023) approach to model diverse user preferences more effectively. *MOLMA* (Zhang, 2023) explores how to balance conflicting multi-objective preferences in language model alignment through reinforcement learning. Both Jang et al. (2023) and Li et al. (2024c) focus on personalizing language models to generate content tailored to individual users, extending alignment objectives from general preferences to personal objectives. These works aim to push the boundaries of multi-objective alignment by exploring strategies to align LLMs with the unique needs of different users.

**Prompt-driven Preference Generation**: Due to the complexity and overhead associated with training multiple reward models or policies for multi-objective alignment, there is growing interest in developing methods that enable preference control using a single model. Yang et al. (2024) propose *Rewards in Context* (RiC), a supervised training approach that incorporates preference tags into prompts, demonstrating superior multi-objective control compared to non-prompt-based methods like MORLHF and Rewarded Soups. Guo et al. (2024b) introduce a flexible prompt-based paradigm, arguing that it is unnecessary to seek the Pareto front every time during generation, as real-world applications rarely require satisfying all preferences simultaneously. Their method prioritizes a single preference while disregarding others and further extends preference control to intensity levels, allowing for more granular customization. Lee et al. (2024) present *Janus*, a framework that aligns with various preferences through system prompts, offering a comprehensive set of preference categories supported by the Janus model.

## 5 CONCLUSION

In this work, we propose MOSLIM, a novel framework for multi-objective alignment in Large Language Models (LLMs) that leverages a single reward model and policy model to efficiently satisfies diverse human preferences. MOSLIM significantly reduces training complexity and resource requirements and eliminats the need for preference-specific supervised fine-tuning (SFT), which enables the use of off-the-shelf models. Experimental results on various multi-objective benchmarks demonstrate that MOSLIM outperforms existing approaches in terms of preference controllability, robustness, and computational efficiency. Moreover, our method provides a flexible and scalable solution for aligning LLMs with complex preference combinations, offering fine-grained control over both dimensions and intensities. This work advances the state-of-the-art in preference-based LLM alignment and opens new avenues for research into personalized and dynamic preference optimization in large language models.

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

# A  INPUT OF REWARD MODEL AND POLICY MODEL

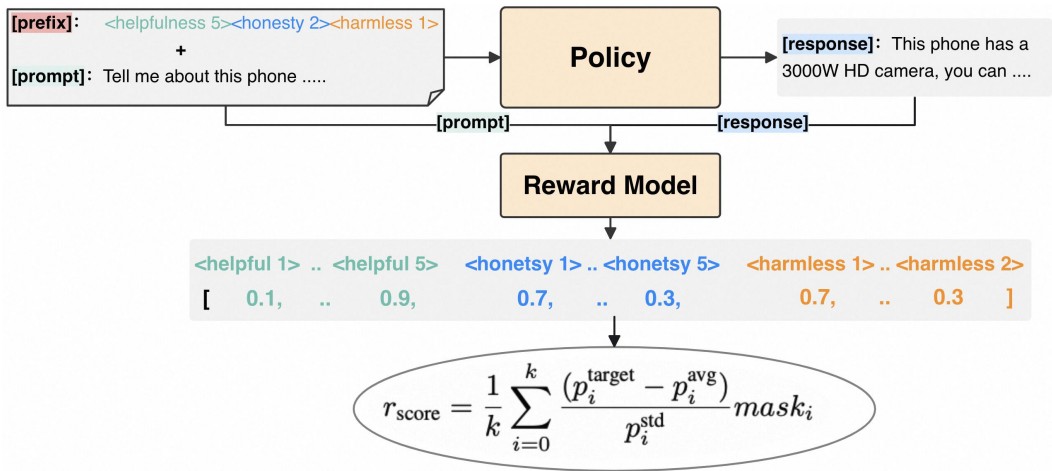

Figure 7: Examples of input for policy and reward model

# B  DETAILED DATATYPE DEFINITIONS

**Definition of *DataType*** Specifically, we define three types of preferences: *preference* $\in$ {*helpfulness*, *honesty*, *harmless*}, resulting in three distinct preference dimensions. For the intensity dimension, the original score distribution in the UltraFeedback dataset ranges from 1 to 5, while in the UltraSafety dataset, the scores range from 0 to 1. To standardize these scores, we define different preference intensity ranges and partition the data accordingly. In the Table 5 in Appendix B, $<preference\ n>$ represents a preference intensity of $n$ ($1 \leq n < n_{max}$), where a larger $n$ indicates a higher intensity. Finally, based on different values of $n_{max}$, we categorize the data into four types. In *DataType* 4, we maintain the highest level of granularity, while in *DataType* 1, the preference dimension is simplified to two categories, removing the need to predict preference intensity and focusing solely on preference classification.

Table 5: Overview of category types and levels for each data type used in the experiments.

| Data Type | Category Description | Categories |
|---|---|---|
| *DataType 1* | `<helpfulness 1>`
`<honesty 1>`
`<harmless 1>` to `<harmless 2>` | 4 |
| *DataType 2* | `<helpfulness 1>` to `<helpfulness 2>`
`<honesty 1>` to `<honesty 2>`
`<harmless 1>` to `<harmless 2>` | 8 |
| *DataType 3* | `<helpfulness 1>` to `<helpfulness 3>`
`<honesty 1>` to `<honesty 3>`
`<harmless 1>` to `<harmless 2>` | 18 |
| *DataType 4* | `<helpfulness 1>` to `<helpfulness 5>`
`<honesty 1>` to `<honesty 5>`
`<harmless 1>` to `<harmless 2>` | 50 |

## C    CONSTRUCTION OF DATASETS

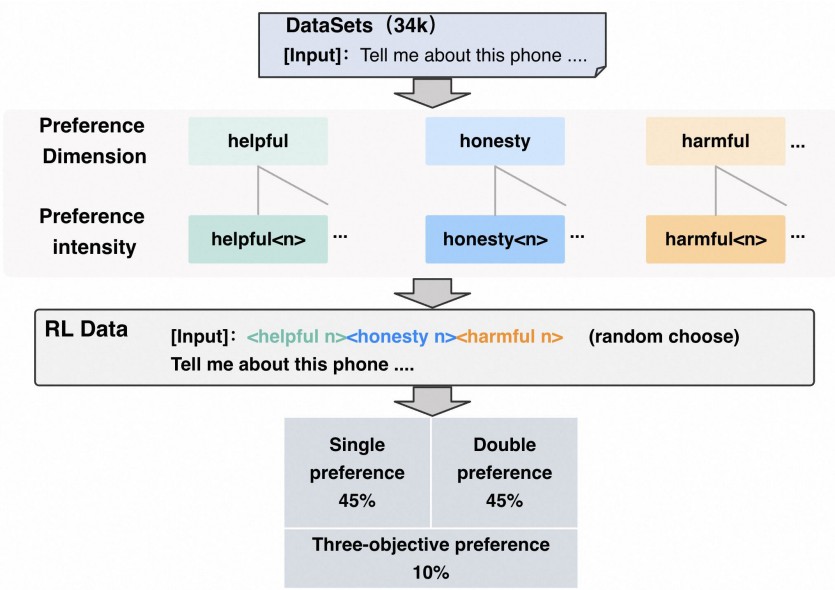

Figure 8: Construction process of reinforcement learning datasets. Each sample consists of a preference prefix and a prompt, where the prefix sets the target preference for the task.

# D HYPERPARAMETERS

Table 6: Parameter Settings for RLHF

| Method | Parameter | Value |
|---|---|---|
| MORLHF | learning_rate | 1e-5 |
| | num_train_epochs | 3 |
| | batch_size | 16 |
| | reward_model_size | 7B |
| | num_reward_models | 3 (one per preference) |
| | optimizer | AdamW |
| RSoup | learning_rate | 2e-5 |
| | num_train_epochs | 4 |
| | batch_size | 32 |
| | reward_model_size | 57B |
| | num_policies | 3 (merged) |
| | optimizer | AdamW |
| MOSLIM$_{PPO}$ | learning_rate | 1e-6 |
| | num_train_epochs | 2 |
| | batch_size | 64 |
| | reward_model_size | 7B |
| | clip_range | 0.2 |
| | entropy_coefficient | 0.01 |
| MOSLIM$_{RLOO}$ | learning_rate | 1e-5 |
| | num_train_epochs | 3 |
| | batch_size | 32 |
| | reward_model_size | 7B |
| | optimizer | AdamW |
| | evaluation_strategy | steps |
| MOSLIM$_{Online\text{-}DPO}$ | learning_rate | 2e-5 |
| | num_train_epochs | 5 |
| | batch_size | 16 |
| | reward_model_size | 72B |
| | clip_range | 0.3 |
| | max_grad_norm | 1.0 |

Table 7: Hyperparameter Settings for Classifier Training

| Parameter | Default Value |
|---|---|
| learning_rate | 1e-5 |
| num_train_epochs | 1 |
| weight_decay | 0.01 |
| batch_size | 32 |
| optimizer | AdamW |
| max_grad_norm | 1.0 |
| scheduler_type | linear |
| warmup_steps | 500 |
| gradient_accumulation_steps | 4 |
| seed | 42 |
| logging_steps | 50 |
| evaluation_strategy | steps |
| save_steps | 1000 |
| fp16 | True |

# E    COMPARATIVE EVALUATION WITH GPT-4

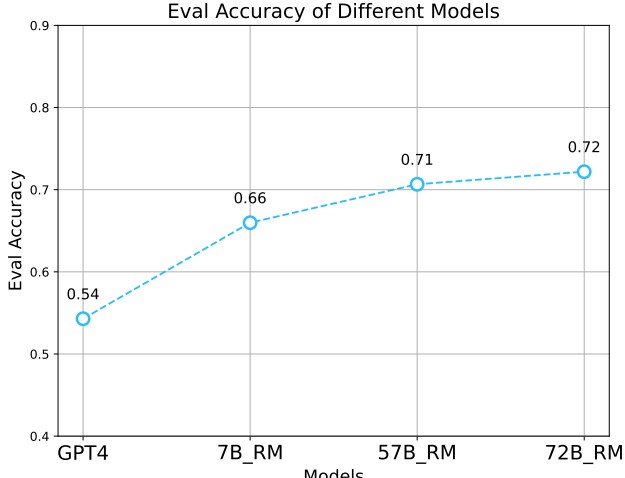

Figure 9: Performance comparison between our reward model and GPT-4.

**Comparative Evaluation with GPT-4** We compare our reward model to GPT-4 using modified evaluation templates derived from standard benchmarks such as MT-Bench Zheng et al. (2023), HaluEval 2.0 Li et al. (2024b), and Hackaprompt Schulhoff et al. (2024). GPT-4 is employed to classify test data, with detailed evaluation protocols provided in the appendix. The experiments are conducted on the *DataType* 2 dataset, and the results are illustrated in Figure 9. The findings reveal substantial performance differences between GPT-4 and our specialized reward model, with a maximum accuracy gap of up to 20% in favor of the *RM-72B*. Even with the smallest model, *RM-7B*, our reward model outperforms GPT-4, achieving up to 15% higher accuracy. These results underscore the effectiveness of our multi-head, multi-label classification reward model in achieving competitive performance.

# F    TRAINING GPU HOURS OF DIFFERENT METHODS

Table 8: Comparison of GPU Hours

| Method | GPU Hours | Model Parameters |
|--------|-----------|------------------|
| MORLHF | 196 | Policy Model: 7B |
|        |     | Reward Model: 7B |
| RewardSoup | 400 | Policy Model: 7B |
|            |     | Reward Model: 7B |
| MOSLIM | **164** | Policy Model: 7B |
|        |         | Reward Model: 7B |
|        |         | Value Model: 7B |

