# OpenReview forum: "MOSLIM:Align with diverse preferences in prompts through reward classification"
_ICLR.cc/2025/Conference — Submitted to ICLR 2025_

### Official Review · Reviewer_sLF4 · 2024-10-31

**Soundness:** 2
**Presentation:** 1
**Contribution:** 2
**Rating:** 3
**Confidence:** 4

**Summary:**

This paper presents MOSLIM, a novel approach to multi-objective alignment of LLMs. The main contribution is to fit the multiple rewards into a single architecture with multiple heads. Then by prompt conditioning, the policy can optimize a linear combinaison of those rewards. At inference time, prompting the network correctly can lead to diverse behaviours, improving over alternatives MORLHF baselines.

**Strengths:**

* Multi-objective RLHF is an area of interest for the ICLR community.
* Using an architecture with multiple heads is a practical approach to combine multiple RMs into a single one.
* Using a classification approach for reward estimation is an interesting direction, notably regarding the recent findings in "Stop Regressing: Training Value Functions via Classification for Scalable Deep RL"?.

**Weaknesses:**

* The paper is very unclear. Some sections are overly complicated to explain simple things such as softmax or cross-entropy. This complexity makes things way more complex that required. For example, rather thansaying "a scalar reward derived from a mapping function that converts classification results from reward model into reward scores" you could just say "combine the multiple rewards by weighted sum".
* Weak contributions. "The prompt-driven mechanism enables flexible adaptation to varying preference intensity combinations" has previously been used. The choice of the reward architecture is not novel and not sufficiently ablated. The reward mapping is actually a weight sum.
* The experiments lack ablations. What the pros and cons of such architecture; do you have transfer across tasks, or in contrast do you lose some?
* Nit. the sum in Equation (11) contains k+1 elements.
* Nit. Multiple typos in Figures: "RSopu" in Fig1, "Multiple Lable" in Fig2, "helpfulness or helpful" in Fig 7...

**Questions:**

* "For computational convenience, we aggregate the losses": could you explain why is it more convenient?
* when you say: "we categorize a question-answer (Q,A) sequence into preferences such as helpfulness, harmlessness, or honesty", do you do it automatically or with human in the loop?
* Could you discuss the connections between your RM classification approach and the paper "Stop Regressing: Training Value Functions via Classification for Scalable Deep RL"?
* How do you explain that Rewarded soups perform better than MORLHF? and actually, which interpolating coefficients did you choose for those methods?

See weaknesses for more questions.

---

### Official Review · Reviewer_YiP8 · 2024-11-03

**Soundness:** 3
**Presentation:** 3
**Contribution:** 2
**Rating:** 5
**Confidence:** 3

**Summary:**

The paper presents MOSLIM, a framework designed to align LLMs with multiple, diverse user preferences through a single reward and policy model. MOSLIM uses a single multi-head reward model and policy model to align with various preference objectives and intensities at training. At inference, it adapts flexibly to different preference intensity combinations using a prompt-driven mechanism. MOSLIM enables efficient, scalable preference alignment, allowing integration with existing off-the-shelf models.

**Strengths:**

- MOSLIM outperforms existing methods such as MORLHF and Rewarded Soups, while achieving controllable alignment across different preference dimensions and intensities.
- The model’s effectiveness is thoroughly validated across several benchmarks (MT-Bench, HaluEval 2.0, Hackaprompt). The study explores various model scales and compares different algorithms (PPO, RLOO, Online-DPO), demonstrating MOSLIM’s robustness across configurations.

**Weaknesses:**

- Contribution and Novelty: The approach of using a multi-head RM for preference alignment has been introduced in prior work [1,2], which may also support multi-objective preference classification. Additionally, the fast inference strategy utilized by MOSLIM shares similarities with previous efforts to dynamically adjust preferences, such as Rewards-in-Context [3]. While MOSLIM mentions that some methods use SFT loss primarily to enhance core abilities (lines 55-62), this claim may overlook the use of rejection sampling for alignment in works like Llama 2 [4] and Gao et al. (2023) [5].
- Explanation for Outperformance of Baselines: The paper does not sufficiently explain why MOSLIM outperforms MORLHF and Rewarded Soups, in terms of the policy performance. It would be beneficial to include both empirical comparisons and theoretical explanations of the different elements of MOSLIM with MORLHF and Rewarded Soups, for example, the performance of RM (here only the model size is compared).
- Representation: Figure 4 may be unnecessary, as it duplicates information already presented in Table 1.

[1] Li, Lihe, et al. "Continual Multi-Objective Reinforcement Learning via Reward Model Rehearsal."
[2] Yang, Adam X., et al. "Bayesian reward models for LLM alignment." arXiv preprint arXiv:2402.13210 (2024).
[3] Yang, Rui, et al. "Rewards-in-context: Multi-objective alignment of foundation models with dynamic preference adjustment." arXiv preprint arXiv:2402.10207 (2024).
[4] Touvron, Hugo, et al. "Llama 2: Open foundation and fine-tuned chat models." arXiv preprint arXiv:2307.09288 (2023).
[5] Gao, Leo, John Schulman, and Jacob Hilton. "Scaling laws for reward model overoptimization." International Conference on Machine Learning. PMLR, 2023.

**Questions:**

Please refer to the Weaknesses part.

---

### Official Review · Reviewer_7z7m · 2024-11-04

**Soundness:** 3
**Presentation:** 2
**Contribution:** 2
**Rating:** 3
**Confidence:** 3

**Summary:**

This paper introduces MOSLIM, a multi-objective alignment framework for Large Language Models (LLMs) that optimizes diverse human preferences through a single reward and policy model. Unlike traditional methods, MOSLIM uses a multi-head reward model for classification-based preference alignment and prompt-driven policy optimization, reducing the need for extensive retraining on specific preferences. The authors validate their approach across multi-objective benchmarks, demonstrating that MOSLIM is more efficient in GPU usage and improves alignment flexibility compared to existing methods like MORLHF and Rewarded Soups.

**Strengths:**

- By employing a single reward model for diverse preferences, MOSLIM significantly reduces computational overhead. This enables off-the-shelf models without requiring fine-tuning for each new preference.
- The framework shows potential scalability, given its ability to handle multiple preference objectives without complex adjustments during training.

**Weaknesses:**

- While MOSLIM addresses multi-objective alignment, the core approach builds on established methods, particularly prompt-driven alignment and multi-head reward models. The contributions are incremental rather than ground-breaking, as the framework primarily refines and consolidates existing techniques.
- The writing is hard to understand, and the word usage is inconsistent in the paper (e.g., Both RSoup and RSopu are used.)
- Limited comparison with a few baselines (Rsoup and MORLHF). Please add more baselines dealing with multi-objective RLHF (RiC [1], RLPHF [2]).
- The paper compares MOSLIM primarily against a few specific methods, but additional comparisons with other state-of-the-art frameworks in multi-objective alignment could provide a more comprehensive evaluation.

---
**References**\
[1] Rui Yang et al., Rewards-in-context: Multi-objective alignment of foundation models with dynamic preference adjustment, ICML 2024.\
[2] Joel Jang et al., Personalized Soups: Personalized Large Language Model Alignment via Post-Hoc Parameter Merging,  https://arxiv.org/pdf/2310.11564

**Questions:**

- Minor: Reward Soups must be abbreviated as RSoup, not Rsopu.
- How does MOSLIM handle cases where preference intensities are not explicitly defined or vary significantly in real-world scenarios?
- Could the authors provide more detail on the decision process for choosing specific intensity levels within the reward mapping function?
- Unlike baseline approaches, the authors have emphasized the advantage of removing SFT in MOSLIM, but there have been no experiments verifying such advantages.

---

### Official Review · Reviewer_HeRK · 2024-11-04

**Soundness:** 2
**Presentation:** 3
**Contribution:** 2
**Rating:** 5
**Confidence:** 2

**Summary:**

This paper proposes a new multi-objective LLM alignment method which trains a single multi-head classification reward model for policy optimization and manipulates diverse preferences through prompting during inference. The method MOSLIM achieves both training and inference efficiency compared to two baselines MORLHF and RSoup. MOSLIM  also enables granularity control over preference intensity level.

**Strengths:**

The paper structure is well-organized and the main concepts are well-articulated. The methodology section is clear and sound with sufficient summary of the previous methods at start. The differences between the proposed method with baselines are explicitly demonstrated through the figures. In the experiments section, the outline at the front is followed by corresponding verification of the experiments results.

**Weaknesses:**

1. Experiment designs may need improvements.
- Although the authors mention the latest work RiC and CDPO, it seems that there are not direct comparison results shown in the manuscript. If feasible, it would be more persuasive  if the authors could show the superiority of MOSLIM over these two methods through the comparisons in the performance or training cost.
- For Figure 5, I am not sure whether it is a fair comparison with two baselines since these are not trained with certain granularity of preference intensities.
2. There are lots of typos throughout the whole manuscript. Please check the grammatical errors carefully. Here are some examples:
- LIne 293, it should be "three SFT models"
- Line 295 296, " is scaling law still consist" and "weather".
- Table 2 it should be "RSoup"
3. Some important details about experiments are missing.
- It could be useful if some details about the evaluation process in Table 2 are added. Do different data types mean different forms of training dataset used in policy optimization? Are you taking the granularity level (i.e. the number of preference intensity classes) into account during evaluation?
- For Table 8, providing time breakdown of reward model and policy model training would make the comparisons more pronounced.

4. Others.
- In equation (10), since the number of classes K could be different for different preference dimension. it is not appropriate to use single K.

**Questions:**

1. In line 250, "during the training phase", is this refer to training of reward models? Is it correct that we need $r_{score}$ for policy training?
2. What does Value Model in Table 8 mean?
3. In table 1, what are the differences between preference accuracy and intensity accuracy.

---

### Meta-Review · Area_Chair_C4zs · 2024-12-14

**Metareview:**

This work introduces MOSLIM, a multi-objective method for aligning large language models (LLMs). The core approach involves training a multi-head reward model and optimizing LLMs using a linear combination of these rewards. While the idea is intriguing, all reviewers have raised substantive concerns regarding the technical soundness of this work, including issues of limited novelty, unclear experimental setups, insufficient results, and unclear writing. Unfortunately, these concerns were not adequately addressed during the discussion period. I am therefore recommending the rejection of this submission.

**Additional Comments On Reviewer Discussion:**

There appear to be widespread and unresolved concerns regarding the soundness of this paper's results but the authors haven't provided any responses to the reviews.

---

### Decision · Program_Chairs · 2025-01-22

Reject